# Experiments on the influence of spot fire and topography interaction on fire rate of spread

**Michael Anthony Storey**[1]*, **Owen F. Price**[1], **Miguel Almeida**[2], **Carlos Ribeiro**[2], **Ross A. Bradstock**[1], **Jason J. Sharples**[3]

**1** Centre for Environmental Risk Management of Bushfires, University of Wollongong, Wollongong, NSW, Australia, **2** Association for the Development of Industrial Aerodynamics, Coimbra, Portugal, **3** School of Physical, Environmental and Mathematical Sciences, University of New South Wales (UNSW), Canberra, ACT, Australia

* mas828@uowmail.edu.au

**Data Availability Statement:** All relevant data are within the manuscript and its Supporting Information files.

## Abstract

Spotting is thought to increase wildfire rate of spread (ROS) and in some cases become the main mechanism for spread. The role of spotting in wildfire spread is controlled by many factors including fire intensity, number of and distance between spot fires, weather, fuel characteristics and topography. Through a set of 30 laboratory fire experiments on a 3 m x 4 m fuel bed, subject to air flow, we explored the influence of manually ignited spot fires (0, 1 or 2), the presence or absence of a model hill and their interaction on combined fire ROS (i.e. ROS incorporating main fire and merged spot fires). During experiments conducted on a flat fuel bed, spot fires (whether 1 or 2) had only a small influence on combined ROS. Slowest combined ROS was recorded when a hill was present and no spot fires were ignited, because the fires crept very slowly downslope and downwind of the hill. This was up to, depending on measurement interval, 5 times slower than ROS in the flat fuel bed experiments. However, ignition of 1 or 2 spot fires (with hill present) greatly increased combined ROS to similar levels as those recorded in the flat fuel bed experiments (depending on spread interval). The effect was strongest on the head fire, where spot fires merged directly with the main fire, but significant increases in off-centre ROS were also detected. Our findings suggest that under certain topographic conditions, spot fires can allow a fire to overcome the low spread potential of downslopes. Current models may underestimate wildfire ROS and fire arrival time in hilly terrain if the influence of spot fires on ROS is not incorporated into predictions.

## Introduction

Spotting has played a major role in some of the largest and most destructive wildfires on record [1–3]. In comparison to a continuous line of fire, spotting during wildfires creates complex discontinuous patterns of fire spread through the ignition of "spot fires". Spot fires are smaller separate fires that ignite after small pieces of flaming or smouldering material (mostly vegetation such as bark or small branches, but also other materials such as building material) are transported by wind (e.g. ambient wind, plume) from the main fire to ignite unburnt fuels [4,

**Funding:** This study is part of the PhD of M.S., for which he received a scholarship from the Bushfire and Natural Hazard Cooperative Research Centre (https://www.bnhcrc.com.au/) and University of Wollongong (https://www.uow.edu.au/). The funders had no role in study design, data collection and analysis, decision to publish, or preparation of the manuscript.

**Competing interests:** The authors have declared that no competing interests exist.

5]. Spotting can easily jump fuel breaks such as roads, which can substantially reduce the chance of a wildfire being contained by fire crews. Short to moderate distance spot fires generally merge with the main fire. This process is thought to increase the rate and in some cases, such as where profuse spotting is occurring, become the main driver of wildfire spread [6–8].

The manner in which a wildfire spreads across a landscape can be complex when spotting is occurring. Wildfires can spread not just as a single fire front but can incorporate spot fires that are generated from and subsequently merge with the main fire front. In this situation, a spot fire could be deemed to increase the rate of spread (ROS) of a wildfire if the coalescence of the spot fire and main fire results in a downwind extension of the wildfire. For example, once the main fire and spot fires merge, the overall downwind extent of the fire front will be the farthest downwind point of the original spot fire(s), which is farther downwind than if the spot fire(s) did not exist, as demonstrated in Fig 1. In this paper, we describe this as a faster "combined" ROS, meaning the overall ROS incorporating the main fire and any merged spot fires (Fig 1). According to this definition, spot fires that are still separate to the main fire would not be considered when measuring combined ROS. This effect has been described as a "jump in fire front position" [9] or, when many spot fires are being generated, that the fire front may "appear to be moving as a continual coalescence of spot fires" [8]. Fire-fire interaction (e.g. increases in radiant heat and pyroconvective feedbacks) [10, 11] and junction zone behaviour (e.g. increased intensity/radiant heat) [12, 13] may enhance spread directly in the area of merging, but may also have wider effects that enhance spread in nearby parts of the fire.

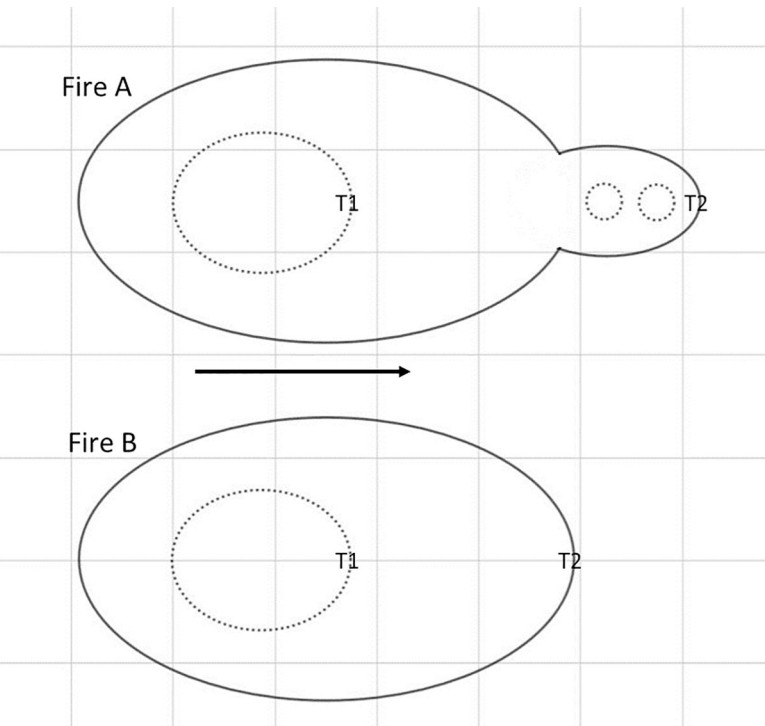

**Fig 1. Simple representation of spread of a spotting fire (Fire A) vs a non spotting fire (Fire B).** Purpose is to demonstrate how spot fire merging may increase combined rate spread (ROS), even with identical main fire ROS. Black arrow is spread direction. Dotted lines are time 1 and solid lines are time 2. At time 1, the fires are the same, but Fire A has two separate spot fires (small circles) and Fire B has none. At time 2, "combined" ROS for Fire A is greater than for Fire B because Fire A includes the two merged spot fires, thus is farther downwind. T1 and T2 indicates head fire location at time 1 and time 2 respectively. Note that the spot fires are not considered when identifying the head fire location for Fire A time 1.

Although spotting may enhance the combined ROS, some spot fires may have little impact or may possibly reduce ROS by removing fuel ahead of the main fire [14, 15]. Numerous factors potentially control how spot fires influence fire spread including main and spot fire ROS (pre-merging), fire intensity and size, number of spot fires, spotting distances [16] and spot fire build-up times [15]. All of these factors are influenced by weather, fuel and topography [7, 17, 18]. Depending on local combinations of these factors, it is likely that similar levels of spotting could significantly influence the combined ROS in some wildfires but not others.

Topographic slope is recognised as an important factor influencing ROS and is usually included as a predictor in ROS models [19, 20]. Typically, compared to level ground, faster ROS is expected for fires travelling upslope and slower ROS expected for fires travelling downslope [21–23]. However, discontinuous spread via spotting has the potential to overcome the effects of slope [7, 24]. This is because firebrands can be transported across areas of low spread potential (downslopes, wet valleys) to ignite new spot fires on the next slope or ridge, which subsequently merge with and extend the main fire (Department of Environment Land Water and Planning unpublished data) [25]. Spotting can also exacerbate instances of dynamic spread driven by interactions between wind, steep slopes and an active fire to produce rapid rates of lateral fire spread along ridges perpendicular to the predominant wind direction [26, 27].

Capturing the effect of spotting is a major limitation of most current operational fire spread models [14]. Although recent work has been done to investigate the most effective way to include spotting in ROS models [25, 28, 29], spread models typically have not considered the influence of spotting on ROS, particularly how spotting and topography (e.g. slope) may interact to influence ROS. For example, McArthur [7] refers only generally to this interaction, by suggesting that when spotting is occurring (on large fires across multiple ridge-valleys) the effect of slope is negligible [20]. A greater understanding is required if the contribution of spotting is to be more accurately reflected in ROS model predictions. As such, there is the need for experimental studies to understand how spot fires and topography interact, and how this influences fire spread.

We conducted a series of small experimental fires in the 6 m x 8 m combustion wind tunnel at the Forest Fire Research Laboratory of ADAI (Association for the Development of Industrial Aerodynamics) in Lousã, Portugal. The experiments incorporated variation in topography and spot fire numbers–specifically, we investigated the following research questions:

- Does the presence of downwind spot fires increase the combined forward ROS of a fire?

- If so, does the effect increase with more spot fires?

- Does any effect of the spot fires depend on the presence or absence of a hill?

- Do spot fires increase the combined ROS both directly (i.e. where spot and main fires merge) and indirectly (offset from where spot and main fires merge)?

## Methods

### Materials and experiment configuration

A set of 30 experiments (Table 1) were conducted over six test days in the TC 3 combustion wind tunnel (Fig 2) at the Forest Fire Research Laboratory of ADAI in Lousã, Portugal. The tunnel is housed in a large semi-open building subject to ambient air conditions (temperature and humidity). The effect of two explanatory variables on combined ROS were investigated with the experiments: number of spot fires (0, 1 or 2) and presence or absence of a model hill (Figs 2, 3 and 4). All experiments were conducted under a constant air flow at 1.5 m s$^{-1}$. Based

**Table 1. Summary of number of experiments performed.**

| Hill | Spot fires | Repetitions |
|---|---|---|
| Hill absent (Flat fuel bed) | 0 | 5 |
|  | 1 | 5 |
|  | 2 | 5 |
| Hill present | 0 | 5 |
|  | 1 | 5 |
|  | 2 | 5 |
| Total experiments |  | 30 |

For all experiments main line fire was ignited and wind speed was 1.5 m s$^{-1}$.

on the experience of the researchers using the combustion wind tunnel for similar fire spread experiments [30], this can be considered a moderate wind speed that produces wind driven surface fires that are suitable for observations of fire spread and merging behaviour.

Dead mature *Pinus pinaster* needles were collected and stored in bulk in the laboratory prior to the experiments. We distributed the same amount of needles for each experiment at an average depth of 4 cm across a 3 m x 4 m (12 m$^2$) area of the combustion wind tunnel's working surface (Figs 3 and 4). For each experiment we used 0.8 kg per m$^2$, or 9.6 kg total, dry weight of fuel, meaning weight of fuel if all moisture was removed (i.e. the dry fibrous needle material only). This dry weight of fuel is above the approximate threshold that allows for continuous fire spread, rather than patchy fire spread, that has been observed in previous experiments in the same laboratory at lower fuel weights [30]. The total fuel weight consists of the dry weight component plus the moisture weight component, where the latter varies with ambient humidity. As we wanted 0.8 kg m$^2$ (i.e. 9.6 kg total) of dry weight for each experiment, we needed to account for changes in ambient humidity affecting the total fuel weight, as measured on a digital scale. We therefore adjusted the total fuel weight collected for each experiment according to its fuel moisture content [12, 21, 30], as measured using an A & D ML-50 moisture analyser. This meant that, for example, when fuel moisture content was lower for a particular experiment, i.e. the moisture weight component was a smaller proportion of total fuel weight, that a lower total fuel weight was required. Over the 30 experiments relative humidity and temperature (measured inside the laboratory) were between 30% and 79% and 19˚C and 29˚C, respectively. Fuel moisture content was between 11.1% and 16% over all experiments, and ranged between 11.1% and 16% for the flat fuel bed experiments (mean 13.7%), and 11.5% and 15.3% for the hill-present experiments (mean 14%).

Fifteen experiments were conducted with a flat fuel bed (hill absent) and 15 experiments had a model hill installed (hill present). For the hill present experiments, the model hill was

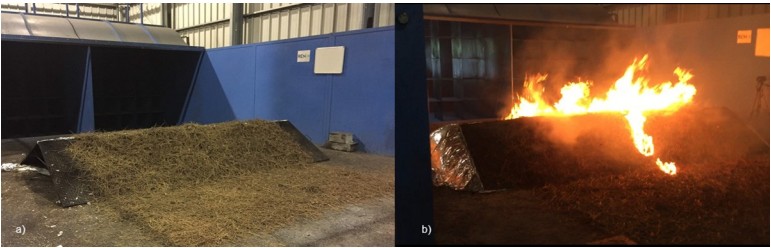

**Fig 2. Images from the TC 3 combustion wind tunnel used for the experiments.** (a) Example of model hill and fuel bed set up (note image was prior to installation of rounded ends on hill). (b) View of model hill (leeward slope) during an experiment with 2 spot fires. Combustion tunnel fan outlets are in the background in both images.

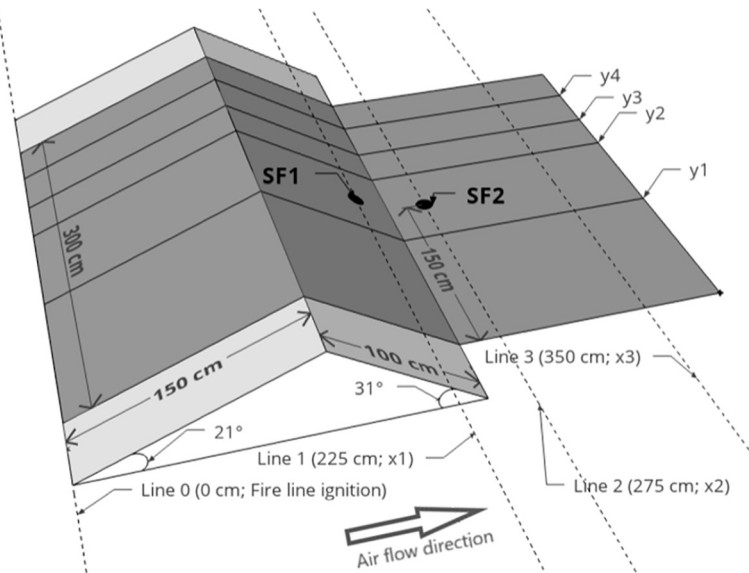

**Fig 3. Schematic representation of experiments with the model hill.** SF1 = Spot fire 1 location, SF2 = Spot fire 2 location. Fuel bed centre y axis (not shown) runs through SF1 and SF2 (i.e. midline between y1 and y2). Grey area is pine needle fuel bed. Head fire was measured between y1 and y2, off-centre fire between y3 and y4. Both head and off-centre fire were measured at three x axis intervals (Line 0—Line 1, Line 0—Line 2, Line 0—Line 3).

400 cm wide (only the centre 300 cm was covered with fuel) and had two slopes: a 150 cm uphill section of slope 21° and a 100 cm downhill section of slope 31° (Fig 3). An additional 150 cm flat section downwind of the hill was covered with fuel (i.e. fuel bed total length 400 cm, total width 300 cm; Fig 3). There were 100 cm gaps between each end of the hill and the combustion tunnel walls. This model hill was used due to is availablity from previous fire behaviour experiments in the laboratory [30]. The model hill itself has no coverings installed at each end (Fig 2A), thus would be open to air flow and the creation of excessive turbulence at the hill ends. To better represent an actual hill, rounded ends were added to smooth the air flow around each end of the hill (Fig 2B). For the hill absent experiments, the fuel bed was simply a 300 cm width x 400 cm length flat surface (Fig 4).

A single fire line was ignited at 0 cm (Figs 3 and 4) using wool thread soaked in a mixture of petrol and diesel in each experiment. The ignition line spanned the entire width of the fuel bed (300 cm). One, two or no spot fires were ignited downwind of the fire line using diesel and petrol-soaked cotton balls placed in the fuel surface. Spot fire 1 and spot fire 2 ignition locations were constant (225 cm and 275 cm downwind) and were both along the centre y axis (Figs 3 and 4). For the hill present experiments, this meant spot fire 1 was 25 cm upslope from the base of the hill (on the leeward slope) and spot fire 2 was on the flat section 25 cm downwind of the base of the hill (Fig 3). The ignition procedure was to: 1) ignite the spot fire(s) cotton balls (when present), 2) allow the spot fire(s) to grow to approximately 10 cm diameter to ensure it was well-established in the fuel bed (a few seconds), 3) ignite the main fire line at 0 cm, 4) turn on the air flow (set to 1.5 m s$^{-1}$ for all experiments) once all needles below the wool thread were ignited, which meant the fire line was well-established in the fuel bed (a few seconds). Only steps 3 and 4 were required for experiments with zero spot fires. The experiment was considered ended when any part of the fire front reached the end of the fuel bed.

Fire progression was recorded using a FLIR ThermaCam SC640 infra-red camera and colour video cameras positioned on a raised platform above (off-nadir) the fires and a side-on colour video camera.

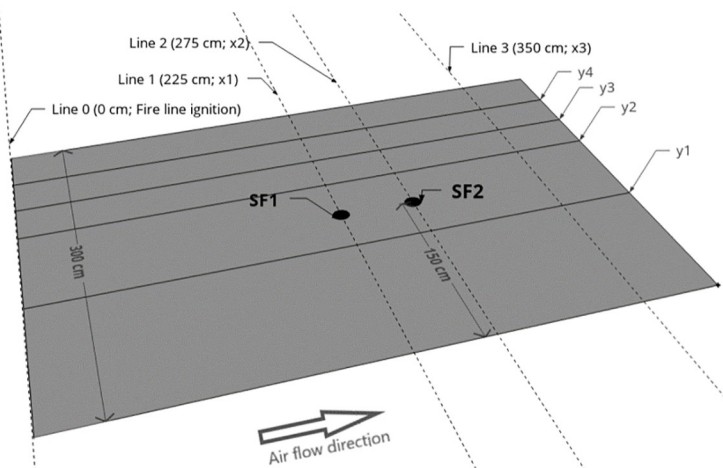

**Fig 4. Schematic representation of the flat fuel bed experiments.** SF1 = Spot fire 1 location, SF2 = Spot fire 2 location. Fuel bed centre y axis (not shown) runs through SF1 and SF2 (i.e. midline between y1 and y2). Grey area is pine needle fuel bed. Head fire was measured between y1 and y2, off-centre fire between y3 and y4. Both head and off-centre fire were measured at three x axis intervals (Line 0—Line 1, Line 0—Line 2, Line 0—Line 3).

Experiments were conducted over six days with similar weather conditions in June 2019. Due to the time necessary to install the model hill and pitot tubes (i.e. air flow measurement devices, data not analysed in this paper), flat fuel bed experiments were conducted first (days 1–3), followed by hill present experiments (days 3–6). Experiments were conducted in sets of 3 (0 spot fire, 1 spot fire and 2 spot fire) with order randomized within each set.

To account for variation in fuel moisture content of the needles across the experiments, which can influence ROS, we used a relative Rate of Spread (RROS; denoted $R'$) instead of ROS (denoted $R$) for analysis. To calculate $R'$, reference tests to measure a "basic ROS" (denoted $R_0$ i.e. ROS of fire line spreading on a surface without wind and slope) were conducted during each test session (approx. every 3 experiments) on a 1 m x 1 m flat table covered with 0.8 kg (dry-weight equivalent) of *Pinus pinaster* needles (as in main experiments) without wind. Basic ROS ($R_0$) and combined ROS from the main experiments ($R$) were used to calculate relative Rate of Spread ($R'$):

$$R\prime = \frac{R}{R_0}$$

$R'$ is unitless and indicates how fast a fire in experiment conditions spread relative to a fire with no wind and no slope ($R_0$); For example, $R' = 2$ means fire in the experiment spread two times faster than $R_0$. Note relative rate of spread $R'$ here is identical to NDROS in Raposo [30].

## Analysis

For each experiment we first measured $R$ as the "combined" ROS from the experiment fires, meaning the overall ROS of the fire incorporating the main fire and any merged spot fires. This means that spot fires that were still separate from the main fire line did not impact the our ROS measurements (e.g. Figs 1, 5A and 6A), but once a spot fire merged with the main fire, the merged area was considered as a single new main fire and ROS was measured incorporating the original spot fire (e.g. Figs 1, 5B, 6B and 6C).

Our ROS measurements were based on visual inspection of images that were extracted at 1 second intervals from the FLIR video recordings (see S2 Appendix for examples). We used

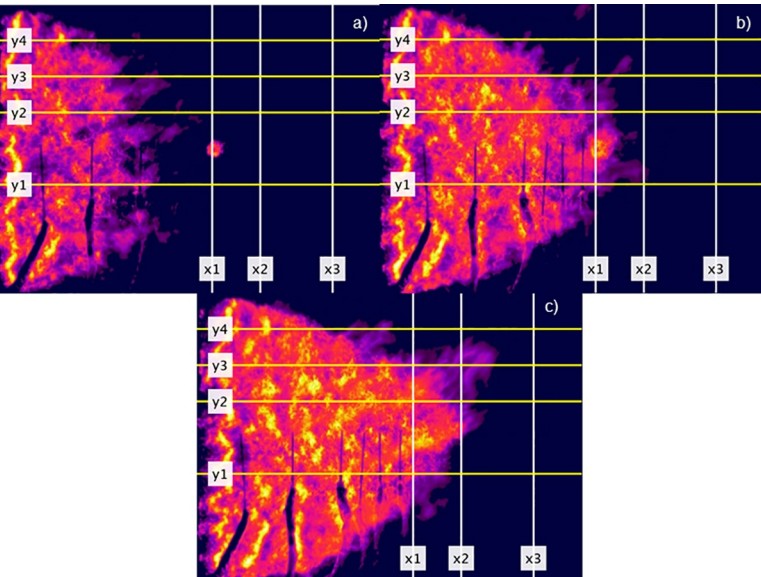

**Fig 5. Series of rectified infrared images extracted from a one spot fire flat fuel bed experiment FLIR recording.**
Head fire was measured between y1 and y2, off-centre fire between y3 and y4. x1 = Line 1, x2 = Line 2, x3 = Line 3.
Images extracted to show examples of (a) before head fire has reached any measurement line or spot fire has merged,
(b) head fire has just crossed Line 1 (x1) and spot fire 1 has merged, (c) head fire has just crossed Line 2 (x2). Seconds
after ignition are (a) 25, (b) 37 and (c) 48.

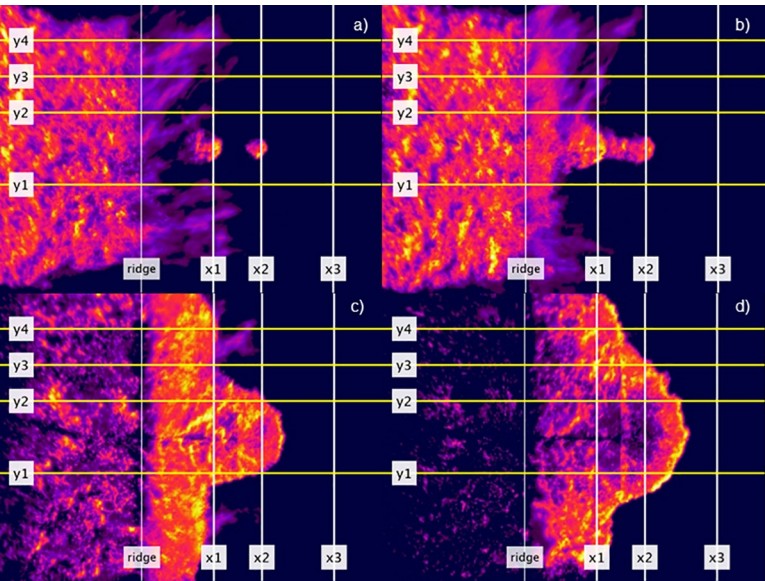

**Fig 6. Series of rectified infrared images extracted from a two spot fire hill-present experiment FLIR recording.**
Head fire was measured between y1 and y2, off-centre fire between y3 and y4. x1 = Line 1, x2 = Line 2, x3 = Line 3.
Images extracted to show examples of (a) head fire approximately at the ridge line and spot fires are still separate (note
light purple area is flame above fuel surface), (b) the main fire and both spot fires have merged, meaning the combined
fire has crossed both Line 1 and Line 2 in the head fire zone, (c) off-centre fire has crossed Line 1 (x1), (d) off-centre
fire has just crossed Line 2 (x2). Seconds after ignition are (a) 27, (b) 39, (c) 117 and (d) 200. Model hill ridge line
shown for reference.

FLIR, as opposed to the colour video, for our analysis as greater discrimination of different flame types was possible with the FLIR (e.g. above surface flame vs burning needles). As the FLIR camera was placed off-nadir, we rectified all images in MATLAB (version R2019a) prior to the visual inspection analysis, by matching a series of control points pairs in the extracted images to a standard flat plane image showing the correct fuel bed dimensions if viewed at nadir (Figs 5 and 6). From the rectified images, we measured the $R$, then calculated $R'$ for three spread intervals along the length of the fuel bed (Figs 3 and 4):

- Line 0 to Line 1 (0 cm to 225 cm, in line with spot fire 1 ignition point)

- Line 0 to Line 2 (0 cm to 275 cm, in line with spot fire 2 ignition point)

- Line 0 to Line 3 (0 cm to 350 cm, 50 cm before the end of the fuel bed)

The Line 0—Line 1 and Line 0—Line 2 intervals were chosen as we considered these best intervals to detect any effect on ROS from one and two spot fires respectively. The Line 0—Line 3 interval was chosen to capture any downwind effects of the spot fire(s). Note there was some small variation (a few centimetres) in the placement of the ignition line (the soaked woollen thread) between experiments, but we do not expect, given that experiments with each unique combination of factors were replicated 5 times, that this substantially affected our results.

We measured both a head fire ROS, within a 75 cm wide zone between y1 and y2 in Figs 3 and 4 (the centre line of the fuel bed was halfway between y1 and y2) and an "off-centre" ROS; a 37.5 cm wide zone between y3 (75 cm from centre line) and y4 (37.5 cm in from edge; Figs 3 and 4). Off-centre ROS was measured only on the side of the fuel bed without pitot tubes and pipes (installed for a separate analysis). This was because there may have been some minor slowing of fire spread on the flank with the pipes, which ran 15 cm above the surface of the fuel bed. This should be noted when interpreting the results, although our observations during the experiments suggested any effect on the spread of the head fire and flank fire on the opposite side to the pipes was negligible. Off-centre ROS was only measured for the Line 0—Line 1 and Line 0—Line 2 intervals because off-centre fire did not reach Line 3 in a sufficient number of experiments. This was because the experiment was ended once the tip of the head fire reached the end of the fuel bed, and in most instances, this was before off-centre fire had reached Line 3.

For each experiment, we visually inspected the images to record the combined ROS (i.e. incorporating main fire and merged spot fires) for each spread interval. Then, to calculate $R'$, the process was to inspect the rectified images, as follows:

1. Record the time at which the entire fire line at Line 0 (0 cm) was ignited (time zero);

2. Beginning within the 75 cm head fire zone, record the elapsed time from time zero to the fire line burning past Line 1, Line 2 and then Line 3. For these 3 spread intervals, time was recorded if pine needles beyond the line (1, 2 or 3) and within the 75 cm head fire zone were ignited. As we measured combined ROS, this meant spot fire(s) only affected recorded times once merged with the main fire (e.g. Fig 6B and 6C); separate un-merged spot-fires did not affect the recorded times (e.g. Fig 6A);

3. Calculate combined ROS using the recorded times and distances for the 3 spread intervals;

4. Calculate $R'$ (i.e. divide combined ROS ($R$) by $R_0$);

5. The process was then repeated for the off-centre fire zone.

Times to cross each line were identified through visual interpretation of the rectified FLIR images, all performed by one person (MS). This required discriminating actual crossings (pine needles ignited across a line) from situations where long flames ahead of the fire (and above the fuel bed) were present in the images. Thus, we first cross-checked the FLIR images with the side-on colour video for a sample of 10 experiments to confirm that above fuel bed flames (lighter purple) and burning needles (reds/yellows; Fig 6) had a sufficiently different appearance in the FLIR images to allow for such discriminations using only the FLIR images.

We conducted two-way Analysis of Variance [31] to test for the effects of hill (presence or absence) and number of spot fires (0, 1 or 2) on R'. R' was log-transformed for the analysis to meet the model assumptions (normality of residuals). A total of 5 two-way ANOVAs were performed; 3 for head fire R' (i.e. R' for the Line 0 –Line 1, Line 0 –Line 2 and Line 0—Line 3 intervals) and 2 for off-centre fire R' (R' for the Line 0 –Line 1 and Line 0—Line 2 intervals). Post-hoc Tukey's Tests [31] and boxplots were used to identify significant differences among treatments (i.e. hill present/absent and spot fire number combinations) and understand the direction of the effects. $p < 0.05$ was our chosen significance level but we also report $p < 0.1$.

## Results

### General observations

Based on general observations of fire behaviour (FLIR, in person, colour videos) during the experiments, there was a large contrast in spot fire behaviour between the hill present and absent experiments. When the hill was absent, the spot fires were still mostly small and circular prior to merging with the main fire front (Fig 5A; Figs B and C in S2 Appendix; S1 Video). However, in all the experiments with spot fires and when the hill was present, spot fires had a larger more elongated shape prior to merging, resulting from clear, predominantly upslope spread, back towards the main fire line (i.e. opposite to main air flow). This is evident in both the FLIR recordings (see S2 Video; Figs E and F in S2 Appendix) and from the side on colour video (Fig 7).

During all the experiments, the spot fires had only a small amount of lateral spread (they remained within the 75 cm head fire measurement zone) prior to merging with the main fire. With the hill present and after merging, there was a clear downwind extension of the new combined fire, particularly for the hill-present 2 spot fire experiments (e.g. Fig 6; Fig F in S2 Appendix; S2 Video).

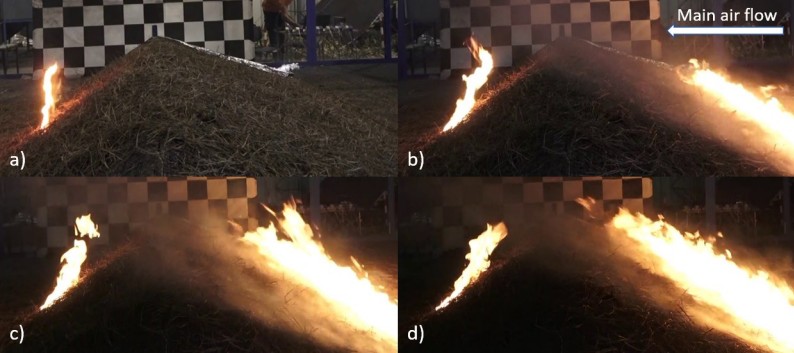

**Fig 7. Still images taken from side-on RGB video camera recording.** Stills are from a one spot fire hill-present experiment, and show how a spot fire (left in each image) spread back upslope, opposite to main air flow, prior to merging with the main fire line (right in each image), which occurred consistently during the experiments hill-present experiments with spot fires. a) shows spot fire just after ignition and prior to fan being switched on and b, c and d) are after fan was switched on and 12, 15 and 18 seconds (respectively) after ignition of the main fire line.

**Table 2. Summary of relative rate of spread $R'$ values for experiments for head fire and off-centre fire at the three spread intervals (Line 0—Line 1, Line 0—Line 2, Line 0—Line 3).**

| Fire type | Hill | Spots | Line 0 –Line 1 | Line 0 –Line 2 | Line 0 –Line 3 |
|---|---|---|---|---|---|
| Head fire | Absent | 0 | 21.8 (15.3–27.9, 5.6) | 19.8 (13.5–27.1, 6) | 16.1 (11.5–21.4, 4.5) |
| | Absent | 1 | 22.2 (17–26.4, 3.4) | 19.7 (14.7–23.9, 3.4) | 15.3 (11.9–20, 2.9) |
| | Absent | 2 | 24.5 (19.8–28.6, 3.7) | 26.7 (20.8–34, 5.5) | 18.6 (13.1–25.3, 4.7) |
| | Present | 0 | 4.6 (3.7–6, 1) | 2.9 (2.4–3.5, 0.4) | 2.3 (2–2.6, 0.2) |
| | Present | 1 | 23.9 (18.6–31.1, 5.8) | 5.7 (4–6.6, 1) | 2.6 (2–3, 0.4) |
| | Present | 2 | 27.3 (22.9–31.7, 3.2) | 25.7 (23.6–27.9, 1.7) | 4.1 (3.6–5.1, 0.6) |
| Off-centre fire | Absent | 0 | 14.8 (8.7–21.7, 5.8) | 12.6 (8.4–17.5, 4.3) | - |
| | Absent | 1 | 13.2 (10.2–15.7, 2) | 11.3 (8.9–13.9, 1.8) | - |
| | Absent | 2 | 14.7 (11.4–21.6, 4.1) | 13.1 (9.6–19.3, 3.8) | - |
| | Present | 0 | 5.3 (4.1–6.8, 1.2) | 3 (2.5–3.7, 0.5) | - |
| | Present | 1 | 6.2 (4.3–8.6, 1.7) | 3.2 (2.7–3.8, 0.5) | - |
| | Present | 2 | 9.8 (7.6–11.6, 1.8) | 5.2 (4.4–6, 0.7) | - |

Values are mean (min-max, sd). A $R'$ value of 20, for example, equates to the experiments having 20 times the ROS of the basic ROS ($R_0$: ROS no wind and no slope). See S1 Appendix for results in raw times (seconds).

In all experiments, the off-centre fire crossed Line 1 after spot fire merging had occurred. However, the first part of the new combined fire to cross Line 1 (in the off-centre fire zone) was from the direct downwind progression of the original main fire line (e.g. Fig 6C). This was also the case for off-centre fire at Line 2 for zero spot fire hill-present and all flat fuel bed experiments. However, for the 2 spot fire hill-present experiments, the first part of the fire to cross Line 2 in the off-centre measurement zone had spread outwards from the original spot fire 2 burning area (e.g Fig 6D). There was occasionally a similar, but less substantial, effect in the 1 spot fire hill-present experiments.

## Head fire $R'$

For the flat fuel bed experiments (hill absent), across all spread intervals measured, mean head fire $R'$ ranged between 15.3 and 26.7 (i.e. 15.3 to 26.7 times faster than $R_0$). In contrast, mean head fire $R'$ for the hill present experiments ranged much more widely, between 2.3 and 27.3 (Table 2). For each unique set of conditions, head fire $R'$ was generally fastest when measured over the Line 0 –Line 1 interval, and slowest over the Line 0 –Line 3 interval, indicating some deceleration in fire spread across the length of the fuel bed.

**Line 0 to Line 1 head fire $R'$.** Two-way ANOVAs indicated that for the Line 0—Line 1 spread interval, head fire $R'$ was affected by a significant ($p < 0.05$) interaction between hill presence and spot fire number (Fig 8, Table 3).

The zero spot fire hill-present experiments (mean 4.6) had the slowest $R'$ which, despite the fast initial upslope spread, was significantly slower (using Tukey's tests with significance level $p < 0.05$) than the zero spot fire flat fuel bed experiments (mean 21.8; Tables 2 and 4). However, the introduction of spot fire(s) overcame the overall slowing effect of the hill; for 1 or 2 spot fire hill-present experiments, $R'$ was similar to the flat fuel bed experiments. Amongst the hill present experiments, $R'$ from the 2 spot fire (mean 27.3) and 1 spot fire (mean 23.9) experiments were significantly faster than the 0 spot fire experiments (mean 4.6), although not significantly different from each other.

Amongst the flat fuel bed experiments, despite some small mean increases with more spot fires ignited, no significant difference in $R'$ as a function of the number of spot fires was detected with the Tukey's tests (means ranged between 21.8 to 24.5).

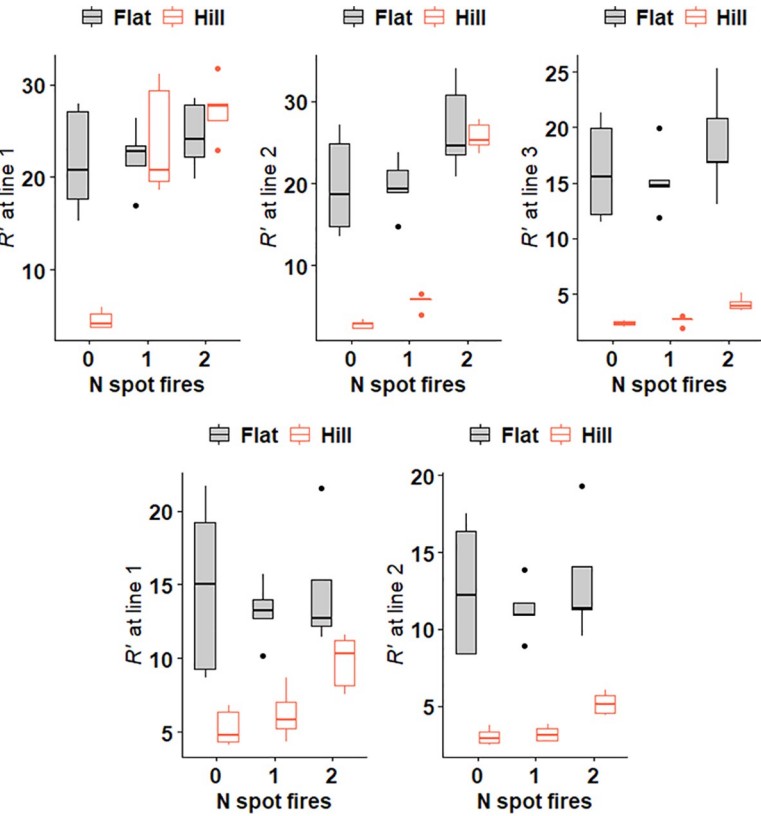

**Fig 8. Boxplots summarising relative rate of spread *R'* from experiments.** Top row is 3 spread intervals analysed for head fire and bottom row is 2 spread intervals for off-centre fire. *R'* grouped by number of spot fires (x axis) and hill present (red boxes) or absent (black boxes grey fill). See S1 Appendix for boxplots summarising raw times.

**Line 0 to Line 2 head fire *R'*.** Two-way ANOVA indicated that head fire *R'* was significantly ($p < 0.05$) affected by an interaction between hill presence and spot fire number (Fig 8, Table 3).

For the flat fuel bed, there was substantial variation of *R'* within spot fire number groups (e.g. *R'* ranged 13.5 to 27.1 for zero spot fires, in raw times between 44 and 72 seconds;

**Table 3. Summary of two-way ANOVA results for effect of spot fire number, slope and interaction term on relative rate of spread *R'*.**

| Interval | Term | d.f. | Head fire | | Off-centre fire | |
|---|---|---|---|---|---|---|
| | | | F | *p* | F | *p* |
| Line 0—Line 1 | Spots | 2 | 71.1 | **< 0.001** | 4.3 | **0.024** |
| | Slope | 1 | 40.6 | **< 0.001** | 53.7 | **< 0.001** |
| | Spots x Slope | 2 | 57.4 | **< 0.001** | 3.1 | 0.061 |
| Line 0—Line 2 | Spots | 2 | 110.4 | **< 0.001** | 6.4 | **0.006** |
| | Slope | 1 | 217.9 | **< 0.001** | 216.7 | **< 0.001** |
| | Spots x Slope | 2 | 59.1 | **< 0.001** | 3.2 | 0.058 |
| Line 0—Line 3 | Spots | 2 | 9.7 | **< 0.001** | | |
| | Slope | 1 | 566.9 | **< 0.001** | | |
| | Spots x Slope | 2 | 2.8 | 0.080 | | |

*R'* was log-transformed for analysis. Results are separate for head fire and off-centre fire and the 3 spread intervals. d.f. = degrees of freedom, F = F statistic, p = probability. Significant p ($p < 0.05$) in bold text.

**Table 4. Results of post-hoc Tukey's tests, separated for the 3 spread intervals and for head fire and off-centre fire.**

| Comparison N spots:hill or flat | Head fire | | | Off-centre fire | |
|---|---|---|---|---|---|
| | p value Line 0—Line 1 | p value Line 0—Line 2 | p value Line 0—Line 3 | p value Line 0—Line 1 | p value Line 0—Line 2 |
| 1:Flat-0:Flat | 1 | 1 | 1 | 0.999 | 0.995 |
| 2:Flat-0:Flat | 0.878 | 0.136 | 0.839 | 1 | 0.998 |
| 0:Hill-0:Flat | **< 0.001** | **< 0.001** | **< 0.001** | **< 0.001** | **< 0.001** |
| 1:Hill-0:Flat | 0.967 | **< 0.001** | **< 0.001** | **0.001** | **< 0.001** |
| 2:Hill-0:Flat | 0.378 | 0.192 | **< 0.001** | 0.302 | **< 0.001** |
| 2:Flat-1:Flat | 0.963 | 0.185 | 0.690 | 0.994 | 0.931 |
| 0:Hill-1:Flat | **< 0.001** | **< 0.001** | **< 0.001** | **< 0.001** | **< 0.001** |
| 1:Hill-1:Flat | 0.996 | **< 0.001** | **< 0.001** | **0.002** | **< 0.001** |
| 2:Hill-1:Flat | 0.545 | 0.256 | **< 0.001** | 0.489 | **< 0.001** |
| 0:Hill-2:Flat | **< 0.001** | **< 0.001** | **< 0.001** | **< 0.001** | **< 0.001** |
| 1:Hill-2:Flat | 1 | **< 0.001** | **< 0.001** | **< 0.001** | **< 0.001** |
| 2:Hill-2:Flat | 0.946 | 1 | **< 0.001** | 0.225 | **< 0.001** |
| 1:Hill-0:Hill | **< 0.001** | **< 0.001** | 0.940 | 0.938 | 0.999 |
| 2:Hill-0:Hill | **< 0.001** | **< 0.001** | **0.002** | **0.013** | **0.008** |
| 2:Hill-1:Hill | 0.833 | **< 0.001** | **0.016** | 0.099 | **0.020** |

p values are for comparisons of mean relative rate of spread *R'* between each pair of experiments configurations of number of spot fires and hill presence or absence (flat fuel bed). Statistically significant values (p < 0.05) in bold.

S1 Appendix). However, despite some evidence of an increase in mean *R'* with 2 spot fires ignited (26.7 compared to 19.7 for 1 spot fire experiments), no significant differences between spot fire levels (0, 1 or 2) were detected with the Tukey's tests (Table 4).

For hill-present experiments, *R'* depended on the number of spot fires that were ignited. Zero spot fire experiments had the slowest *R'* (mean 2.9). *R'* increased significantly with 1 spot fire (mean 5.7) and again with 2 spot fires (mean 25.7). *R'* from the 2 spot fire experiments represented an increase by a factor of 5 and a factor of 9 compared to the 1 spot fire and 0 spot fire experiments, respectively. These results were similar to Line 0—Line 1 results, except now 1 spot fire *R'* was slower than 2 spot fire *R'* when the hill was present.

The results indicated that the ignition of 2 spot fires could overcome the overall slowing effect of the hill. When there were zero spot fires, hill-present *R'* was significantly slower than flat fuel bed *R'*. However, when there were 2 spot fires, hill-present *R'* was not significantly different to *R'* from the 2 spot fire flat fuel bed experiments.

**Line 0 to Line 3 head fire *R'*.** For the Line 0 to Line 3 interval, two-way ANOVAs indicated that head fire *R'* was significantly (p < 0.05) affected by hill presence and spot fire number (Fig 8, Table 3). The interaction between these factors was influential (p < 0.1). While for the Line 0—Line 1 and Line 0—Line 2 intervals spot fire(s) were able to overcome the slowing effect of the hill, this disappeared for the Line 0—Line 3 interval. Tukey's test and boxplots showed significantly slower *R'* for hill-present experiments compared to flat fuel bed experiment regardless of spot fire number (Table 4, Fig 8).

These results reflected the slow spread observed during the experiments on the flat area downwind of the model hill. However, there was still a significant increase in *R'* resulting from downstream effect of spot fires for the hill-present experiments. *R'* from the 2 spot fire hill-present experiments (mean 4.1) was significantly faster than *R'* from both the 0 and 1 spot fire (means 2.3 and 2.6). In raw times, this equates to the 2 spot fire experiments reaching Line 3 around 4 minutes earlier than the 1 spot fire experiments (S1 Appendix). There was no significant effect of spot fire number on *R'* for the flat fuel bed experiments.

### Off-centre fire *R'*

Mean off-centre fire *R'* for the flat fuel bed experiments was similar between spot fire levels, ranging between 11.3 and 14.8 (Table 2). For the hill-present experiments, mean off-centre fire *R'* was slower but ranged more widely; between 3 and 9.8.

**Line 0—Line 1 off-centre fire *R'*.**  Off-centre fire *R'* from Line 0—Line 1 differed significantly as a result of hill presence and spot fire number (Table 3). The interaction between these factors was influential (p < 0.1). Tukey's Tests (using significance level p < 0.05) and boxplots indicated no significant difference in *R'* depending on spot fire number for the flat fuel bed experiments (means 13.2 to 14.8; Tables 2 and 4; Fig 8). For the hill present experiments, 2 spot fire experiment *R'* (mean 9.8) was significantly faster than 0 spot fire experiment *R'* (mean 5.3), and also not significantly different to *R'* for the flat fuel bed experiments (means 13.2 to 14.8). The *R'* for 1 spot fire (mean 6.2) and zero spot fire (mean 5.3) hill present experiments were both significantly slower than all the flat fuel bed experiments.

**Line 0—Line 2 off-centre fire *R'*.**  Similar results were found for the Line 0—Line 2 spread interval. Off-centre fire *R'* was significantly (p < 0.05) affected by hill presence and spot fire number (Table 3), though the interaction between these factors was influential (p < 0.1). The Tukey's Tests and boxplots indicated no significant *R'* difference depending on spot fire number for the flat fuel bed experiments (means 11.3 to 13.1; Tables 2 and 4; Fig 8). However, for the hill present experiments, *R'* from the 2 spot fire experiments (mean 5.2) was significantly faster than both 1 spot fire (mean 3.2) and 0 spot fire experiments (mean 3). Similar to the Line 0—Line 1 results, 2 spot fires were required to create a significant increase in *R'*. Regardless of spot fire number, off-centre fire *R'* was significantly slower for the hill experiments compared to the flat fuel bed experiments.

## Discussion

Our results provide strong evidence that topography can be a major factor controlling how spot fires contribute to fire spread. In our experiments, spot fires had no significant impact on relative rate of spread (*R'*) when the fuel bed was flat, but significantly increased *R'* when a hill was present. The effects were strongest on the head fire, but a significant increase in off-centre fire *R'* was also detected (with 2 spot fires present). The largest effect was for the Line 0—Line 2 interval when the hill was present, where 2 spot fires increased head fire *R'* by a factor of nine compared to zero spot fires.

The large contrast in the results between the flat fuel bed and hill present experiments are likely due to differences in air flow, slope, fire-fire interaction and spot fire growth prior to merging. During the flat fuel bed experiments, with no influence of a hill and a unidirectional air flow, spot fires were overrun by the main fire before they could grow large enough to significantly influence head fire or the off-centre fire *R'*.

In contrast, the model hill may have created areas of high (up windward slope) and low (down leeward slope) spread potential through combined effects of slope angle and winds. In the absence of spot fires, and despite fast initial upslope (up windward slope) spread, the very slow spread downslope (down leeward slope) and downwind of the hill resulted in very low head and off-centre fire *R'* for the 3 spread intervals. However, when spot fire(s) were introduced, they were observed to predominantly spread up the leeward slope toward the ridge, which was opposite direction to main air flow, to merge with the main fire. This meant that what were initially separate fires (main fire and spot fire(s)) became one main fire, with the new head fire front extending to the farthest downwind point of the original spot fire (e.g. Fig F (panels b and c) in S2 Appendix). Given that our measurements were derived from the "combined" ROS, the result was a higher *R'*, particularly for Line 0—Line 1 and Line 0—Line 2

spread intervals. While fires are generally expected to travel faster upslope, other factors are likely to have contributed to this observation. The behaviour may have partially been the result of fire-fire interaction, where the smaller spot fire(s) were drawn towards the larger line fire due to indraft effects [10]. The behaviour may also be explained by the possible formation of a lee-slope rotor, rather than a laminar air flow: i.e. as air flow separated from the surface once past the ridge, a circulation of air that flowed from the base to the top of the lee slope may formed, similar to what has been described in previous experiments in the same laboratory with similar ridge and wind conditions [30]. Although we have not analysed the drivers of the upslope spread here, the result is that the combined fire was able to overcome the low spread potential of the downslope: i.e. the spot and main fires merged into a larger combined fire, with the new fire front extending to the farthest downwind point of the original spot fire.

We also found that having at least two spot fires present significantly increased off-centre fire $R'$ in the hill-present experiments (no significant influence of one spot fire was detected). In comparison to head fires, there was a more indirect effect of the spot fires on off-centre fire $R'$ measured for Line 0—Line 1, as the spot fires did not burn directly into the off-centre fire measurement zone prior to off-centre fire crossing Line 1. A possible explanation for this is that fire-fire interaction and merging of spot fires may have generated greater radiant heat, which led to more rapid drying and ignition of nearby fuels (i.e. in the off-centre measurement zone), thus faster $R'$. The faster off-centre fire $R'$ measured for Line 0—Line 2 (hill present and 2 spot fires) likely resulted from the downstream effects of higher $R'$ for Line 0—Line 1, but also from the outward spread of spot fire 2 (into the off-centre fire measurement zone) after it had merged with the main fire (Fig 6 and Fig F in S2 Appendix). In all the 2 spot fire hill present experiments, the fire front had a much more curved shape due to the extension (and combination of downwind and outward spread) caused by the merging of the 2 spot fires (e.g. Fig 6), compared to the no spot fire hill present experiments where the fire front was relatively flat (e.g. Fig D in S2 Appendix). Some junction zone behaviour, where increased spread is observed between two arms of fire, may have also influenced these results [12, 13].

It is important to consider how the results from our experiments, under a narrow set of conditions, may apply to actual wildfires. A main limitation of our study is the small scale, which was necessary to isolate the conditions (wind, slope, fuel etc.) for the study. Although we have identified basic mechanisms, it is unknown exactly how our results would scale to an actual wildfire. For example, during the hill present experiments, flames shot over the ridge and extended (above the fuel bed) past Line 1 (i.e. 75 cm past ridge, above spot fire 1) and above Line 2 in some cases (i.e. 125 cm past ridge, above spot fire 2; see Figs E (panels a & b) and F (panels a & b) in S2 Appendix). This would have resulted in direct heat exchange with the spot fires that may have enhanced spread and intensity. This may occur in some smaller wildfires in certain topography, but it is unlikely that flames would shoot above spot fires once fires reach a certain size/spotting distance (e.g. a spotting distance of say 100 m would require flames of ∼ 100 m). However, some supplementary experiments we conducted with only one spot fire (no fire line) found that a single spot fire still predominantly spread back upslope, suggesting that such heat exchange was not the major determinant of spot fire spread during the experiments.

The role of spotting in the spread of wildfires is likely controlled by a wider range of factors than those considered in our experiments. These include main fire ROS, spot fire build-up time, distance between spot fire and main fire [15] and fire sizes/intensities, which vary substantially in actual fires. Wildfires also commonly burn across complex topography that includes variable slope angles, multiple ridges and variable orientation of wind to local topographic features.

However, the effects and drivers described above for the hill present experiments are thought to be important in actual wildfires burning in areas with steep slopes. For example,

lee-slope rotors can encompass most of the lee slope in fire prone areas [32], and modelling and observation confirm they are important to the extreme fire behaviour of vorticity-driven lateral spread [26, 27, 33]. Coupled modelling of larger scale hills suggest that spot fires on a lee slope exhibit very similar behaviour of upslope spread as occurred with the experiment spot fires [27, 33]. While lee-slope rotors are important, other wind flows may result in a different pattern of spot fire behaviour and merging, such as more laminar flows resulting in downslope winds (e.g. Foehn winds, katabatic winds) or wind flow of different orientation to the ridge [34, 35].

Our experiments demonstrated that, for some spread intervals, increasing spot fires from one to two can increase combined ROS. It could be reasonably expected that further increasing the number of spot fires (or perhaps spot fire build-up time and size) could further increase combined ROS. Wildfires can produce hundreds of individual spot fires at any one time, including mass spotting events [1, 36, 37] that could cause a fire to, as suggested by Cheney [8]; "appear to be moving as a continual coalescence of spot fires".

Unpublished data (Victorian Department of Environment, Land, Water and Planning— FLIR video and line scans) suggest that during the Wye River wildfire in 2015 (Victoria, Australia), a rapid downslope spread of ~2 km h$^{-1}$ occurred, resulting from a fire on a ridge. This fire set off the ignition of hundreds of spot fires (including spot to spot ignitions) downslope that subsequently merged together, thus creating a new main fire that extended from the original ridge fire to the base of the hill around 2.5 km away. Although infrared video from that fire suggests lee slope rotors may not have formed, the high numbers of spot fires that merged resulted in a similar effect to that of our hill present experiments. Moreover, coupled modelling by Toivanen, Engel [38] found that the spread of the Kilmore East wildfire in 2009 (Victoria, Australia) could be reproduced only if the ignition and merging of 18 downwind spot fires was incorporated. These suggest that, considering the combined ROS, spotting and merging might be important not just over a single slope (as in our experiments), but over a much larger scale (spotting occurred up to 30 km away in the Kilmore East wildfire [1]).

Further research is needed to explore these issues and to further investigate the role of spot fires in wildfire spread. A mixture of experiments, fire spread simulation and empirical wildfire analysis could be used to research, for example, the effect of more complex terrain (e.g. multiple hills, different slope steepness and orientation to wind), different spot fire numbers/ locations and different wind speeds.

Our results have important implications for wildfire spread modelling. In particular, our results suggest spotting can lead to a substantial increase in ROS despite the presence of a downward slope, resulting from a pattern of spot fire and main fire merging. Such a pattern of fire spread has been referenced in general in the literature [7] but has typically not been included in models of ROS. Some models are attempting to incorporate the effects of spotting on ROS; e.g. through stochastic processes [25, 28, 29]. However, our results suggest that the interaction between spotting and topography, where the effect of spotting on ROS varies depending on local topography, is highly influential on fire spread. Our results also suggest that a wildfire's ROS in hilly areas cannot be modelled accurately unless spotting is accounted for implicitly (e.g. incorporating spotting wildfires in the development of empirical spread models) or explicitly (e.g. physical modelling of main and spot fires and patterns of merging).

## Conclusion

Our experiments confirmed that topography can play a substantial role in determining how spot fires contribute to the combined ROS of a fire: the combined presence of a hill and spot fires can potentially elevate ROS. Spot fires may allow head fires to overcome areas of low spread potential (downslope, sheltered area downwind of hill) through merging, along with wider effects (e.g.

possible increased radiant heat) that increase off-centre fire ROS. Predictions of fire spread in a hilly landscape using models that do not incorporate the interaction between spot fire and topography may underestimate combined ROS and fire arrival time. Larger scale experiments (with wider range of conditions), empirical analysis of wildfires and coupled modelling of spread could be used to further understand the contribution of spotting to combined ROS.

## Supporting information

**S1 Appendix.**
(DOCX)

**S2 Appendix.**
(DOCX)

**S1 Video. Example of a two spot fire hill absent experiment FLIR recording.**
(MP4)

**S2 Video. Example of a two spot fire hill present experiment FLIR recording.**
(MP4)

**S1 File. Experiment data for head fire analysis.**
(CSV)

**S2 File. Experiment data for off-centre fire analysis.**
(CSV)

## Acknowledgments

Thanks to Jorge Raposo, Nuno Luís and Gonçalo Rosa that participated in the experiments.

We would like to thank the FCT-Foundation for Science and Technology for the PhD of Carlos Ribeiro with the reference (SFRH/BD/140923/2018). The provision of a PhD scholarship to Michael Storey from the Bushfire and Natural Hazards Cooperative Research Centre and University of Wollongong is gratefully acknowledged.

## Author Contributions

**Conceptualization:** Michael Anthony Storey, Owen F. Price.

**Data curation:** Michael Anthony Storey, Carlos Ribeiro.

**Formal analysis:** Michael Anthony Storey.

**Methodology:** Michael Anthony Storey, Owen F. Price, Miguel Almeida, Carlos Ribeiro.

**Project administration:** Michael Anthony Storey, Owen F. Price, Miguel Almeida.

**Supervision:** Owen F. Price, Miguel Almeida, Ross A. Bradstock, Jason J. Sharples.

**Writing – original draft:** Michael Anthony Storey.

**Writing – review & editing:** Michael Anthony Storey, Owen F. Price, Miguel Almeida, Carlos Ribeiro, Ross A. Bradstock, Jason J. Sharples.

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
