## [Decision Letter · Decision Letter 0]

9 Oct 2020

PONE-D-20-18346

Experiments on the influence of spot fire and topography interaction on fire rate of spread

PLOS ONE

Dear Dr. Storey,

Thank you for submitting your manuscript to PLOS ONE. After careful consideration, we feel that it has merit but does not fully meet PLOS ONE’s publication criteria as it currently stands. Therefore, we invite you to submit a revised version of the manuscript that addresses the points raised during the review process.

We look forward to receiving your revised manuscript.

Kind regards,

Min Huang

Academic Editor

PLOS ONE

Journal Requirements:

Additional Editor Comments (if provided):

Thank you for your patience while we were waiting for additional reviewer comments. Please respond to all reviewers' comments. Note that Reviewer 3 also included some notes in an attachment.

Reviewers' comments:

Reviewer's Responses to Questions

**Comments to the Author**

1. Is the manuscript technically sound, and do the data support the conclusions?

Reviewer #1: No

Reviewer #2: Yes

Reviewer #3: Yes

2. Has the statistical analysis been performed appropriately and rigorously? 

Reviewer #1: No

Reviewer #2: Yes

Reviewer #3: Yes

3. Have the authors made all data underlying the findings in their manuscript fully available?

Reviewer #1: Yes

Reviewer #2: No

Reviewer #3: Yes

4. Is the manuscript presented in an intelligible fashion and written in standard English?

Reviewer #1: Yes

Reviewer #2: Yes

Reviewer #3: Yes

5. Review Comments to the Author

Reviewer #1: General comments

This paper details experiments investigating the role of spot fires in influencing the behavior of fires burning over broken topography. Small-scale experiments in a laboratory setting are utilized to investigate the effect of a small number of variables. While the topic of this ms is of broad importance in the world of wildland fire science, unfortunately the approach taken by the authors and their treatment of the problem is little more than a cursory exploration of the phase space of possibilities with little regard for implication in the broader context of wildland fire behavior.

Critical here is the experimental design and the concept of ‘combined’ fire rate of spread, both of which indicate a lack of thorough consideration of the problem in developing the research question into a meaningful experiment. It’s not clear if the observation in the field upon which this work is based of wildland fire rates of spread increasing with spotting are the result of merging of spot fires with the main fire, or independent propagation of the spot fires, forming a pseudo-front that then is considered the head of the fire. More often than not, spot fires are overrun by the main fire unless the main fire is held-up for some reason (break in topography or fuel, etc). The approach taken here seems to assume the former option, but if this is the case, then the experimental design seems intended to not understand the key variables driving this interpretation. If the speed of a fire is defined only by its progression of a flaming edge regardless of its speed, then slowly moving but separated point fires may be seen to ‘advance’ the fire once they join but only by the distance of the location of the spots since they don’t spread very far and not their rate of spread. As a result, the location of the spot fire and its speed of connection with the main fire then become the critical variables, neither of which are investigated in this study. Interactions of the spot fires with main fire also do not seem to be of much interest given the method of calculating cumulative ROS at three locations. As it stands, the current experimental design is largely a matter of geometry rather than fire behavior and suggests that there is an optimal distance and time for spot fires to merge and increase your ‘combined’ ROS. In my mind there is a fundamental difference between spot fires that actively spread, advancing the fire area, and spot fires that merge but don’t spread but inadvertently increase ‘combined’ ROS.

Furthermore, the experimental design does not consider a meaningful control against which experiments may be compared. Use of a zero wind, zero slope rate of spread of a fire burning on a 1 m^2 area is not meaningful as in both cases (wind and/or slope) the flame propagation mechanism is different to the zero wind and zero slope case and we know that larger fires spread faster. In this work, the control could be the no-spotting case, with and without a hill.

The use of a single hill with arbitrary slope and height does not provide any insight into the effect of a hill on rate of spread as it is not a variable and raises far too many questions for a publication of this type. Why were the values chosen? Were others considered? Why is the hill asymmetric? Why this asymmetry? Similarly, why 1.5 m/s wind? This is also a very important variable that is not recognized or studied appropriately, as is fuel moisture content. Too much of the methodology appears arbitrary without due consideration of the types and ranges of variables involved. Much of the analysis of results is superficial and inference of results in the discussion and conclusions do not adequately consider the limitations of the study. Furthermore, much of the discussion is filled with speculation without any attempt to isolate or specify probable influence.

The documentation of methods utilized in this study is woefully inadequate with insufficient detail to enable others to repeat your experiments. What was the length of the ignition line? How was the flame front defined in the infra-red? What were the ‘similar weather conditions’ of the experiments? Experiments conducted in sets of threes is not random.

The structure of the paper mixes methods and results throughout. A thorough rethink on the contents would improve the logic and flow of the paper. It is not until deep into the discussion that it becomes clear that the primary mechanism of spot fires influencing fire spread is that they spread backward against the main wind direction, which alters the entire perspective of the experiment, its results and implications. The reliance upon speculation of the presence of a lee-slope rotor further weakens the results.

Other general questions that arose: Why were spots allowed to grow to a predetermined size and what relation was this to the time of ignition of the main fire? Why was the wind turned on only after the main fire was ignited? In reality the spots would be ignited after the main fire reached the ridge. What effect did this method have on results? What was the measurement error of the results? Did the FLIR analysis method change with FMC? In the results, what is the effect of fire spread and what is the effect of geometry? Would the results of one spot fire at spot fire location 2 have produced any different result?

Specific comments

Lay readers you need to define many terms such as spotting at first use. In particular, ‘combined ROS’ needs a robust definition.

L85-86: This statement is a truism.

Table 1: Slope angles should be relative to wind direction (windward = positive, leeward = negative).

L151: Why was 0.8 kg/m^2 selected? It seems rather light in regard to wildland fuel loads. How is it representative?

L155-160: This section is not clear and confusing. Results should be kept for results section.

L162: SI units.

L167: Did the rounded ends work? What effect did they have on the overall experimental set up?

L191: Why not use a control?

L211-213: These are not interval rates of spread but cumulative rates of spread. An interval rate of spread would be L1-L2 and L2-L3. Actual interval rate of spread could make for interesting analysis.

Figure 5 and 6: Put times into the figures.

L239: Show this to be the case.

L244: What sort of confounding effects? Should they be present at all for these experiments if they affected the flow? Did they affect ROS? Show this.

Table 2: These results show that the cumulative rate of spread for all experiments (with the exception of Hill-absent 2-spots decreased across each measurement. Why was this the case? Indicates the fires continued to decelerate and thus were not in equilibrium, even without slope and spot fires. Present standard deviations.

L384: No, it doesn’t. You considered one topography and two fixed points which showed these to not be a major factor in affecting fire behaviour.

L392: Did you measure differences in air flow? Did you quantify fire-fire interactions? Where is the data for spot fire growth prior to merging?

L394: How large did a spot fire need to grow to influence head fire behavior?

L395: Pure speculation.

L401-402: This observation needs to be quantified and presented in results. At what speed did this occur?

L408: How common are these? What happens when they are not present? Show that they were important here and not mere speculation.

L411-412: This is not what happened as it was just a matter of geometry.

L417: Speculation.

L423: Define what you mean by ‘more curved shape’.

L426: Speculation.

L428: Small scale but also narrow set of conditions and variables.

L446-454: Relevance of this paragraph?

L463: Relevance?

L491-493: This is not a new finding.

L497: By how much?

Reviewer #2: This is a review of the manuscript entitled “Experiments on the influence of spot fire and topography interaction on fire rate of spread” submitted to the Journal PLOS ONE. The authors describe a controlled and replicated experiment to better understand interactions between terrain and spot fires on fire rate of spread. I commend the authors on a well-planned and executed experiment. The mechanisms by which spot fires contribute to wildfire spread, and how that is incorporated in larger landscape-level fire simulation models is not well understood and this research helps make incremental steps towards that goal. The limitations of the experiment were well described in the manuscript and future steps forward were also addressed. The main comment I have is that the study captures only one dynamic of fire activity (i.e., ROS), but misses other important aspects of spotting that can’t really be observed at such a small scale. In terms of spotting, the most important thing really isn’t ROS, per se, it is the ability of fires to jump fire lines or other natural features that would otherwise be successful barriers to spread. It is the ability of fires to act as ignition sources and create several flaming fronts that can erupt meters to kilometers from the main fire front. The authors should put their results in the context of broader implications of spotting on fire suppression and fire modeling efforts.

Comments:

I very much like the use of the relative rate of spread as the response variable, which accounts for temporal variation in moisture conditions as opposed to raw rate of spread.

It seems as if there was quite a range of environmental conditions: (Lines 158-160) “Over the 30 experiments fuel moisture content ranged between 11% and 16%, relative humidity and temperature (measured inside the laboratory) were between 30% and 79% and 19oC and 29oC, respectively”. It would be good to see the distributions of these variables across the different replicates. Were the hill experiments hotter and drier consistently than the flat slope runs? I get that the R’ is probably the most important aspect of the combustion spread, but it’s really the variability within treatments that is interesting and hasn’t been explored and looking into the environmental variables could help.

It is likely my misunderstanding of your methods, but it’s not clear how ROS is actually accounted when taking into account the spot fire. For instance, what if the spot fires crosses the end of the fuel bed before the fire started at line 0 reached line 1. How would R’ be calculated then?

For such a controlled experiment there appeared to be high variability in R’ within a treatment (i.e., across replicates, Fig 7, 8). Can this variability be explained by variations in local weather during the runs?

What was the variability in R0? Were there big differences across experimental treatments? Perhaps this could be included in the supplementary material. In general, it would be good to know how much the variability in external weather conditions likely influenced model results.

Line 423: should read “do to the extension”

Reviewer #3: It is a very interesting paper addressing an important issue. The title reflect the content of the manuscript. The methodology and analysis are sound . The weaknesses of the experiment are identified by the authors. The manuscript is well written with clear language.

.

6. PLOS authors have the option to publish the peer review history of their article (what does this mean?). If published, this will include your full peer review and any attached files.

Reviewer #1: No

Reviewer #2: No

Reviewer #3: **Yes: **Fantina Tedim

---

## [Author Response · Author response to Decision Letter 0]

9 Nov 2020

Please see the file uploaded with the file name "Responses_to_reviewer_comments.docx".

---

## [Decision Letter · Decision Letter 1]

23 Dec 2020

Experiments on the influence of spot fire and topography interaction on fire rate of spread

PONE-D-20-18346R1

Dear Dr. Storey,

We’re pleased to inform you that your manuscript has been judged scientifically suitable for publication and will be formally accepted for publication once it meets all outstanding technical requirements.

Kind regards,

Min Huang

Academic Editor

PLOS ONE

Additional Editor Comments (optional):

Reviewer #2 was invited to evaluate the authors' responses to all reviewers' comments on their original submission. Based on Reviewer #2's feedback and Reviewer #3's evaluation last time, this paper has met the minimum requirements for publication in PLOS ONE. Therefore it can be accepted as is. Thanks to all reviewers and authors for their efforts.

Reviewers' comments:

Reviewer's Responses to Questions

**Comments to the Author**

1. If the authors have adequately addressed your comments raised in a previous round of review and you feel that this manuscript is now acceptable for publication, you may indicate that here to bypass the “Comments to the Author” section, enter your conflict of interest statement in the “Confidential to Editor” section, and submit your "Accept" recommendation.

Reviewer #2: All comments have been addressed

2. Is the manuscript technically sound, and do the data support the conclusions?

Reviewer #2: Yes

3. Has the statistical analysis been performed appropriately and rigorously? 

Reviewer #2: Yes

4. Have the authors made all data underlying the findings in their manuscript fully available?

Reviewer #2: Yes

5. Is the manuscript presented in an intelligible fashion and written in standard English?

Reviewer #2: Yes

6. Review Comments to the Author

Reviewer #2: This is the second review for the manuscript “Experiments on the influence of spot fire and topography interaction on fire rate of spread” for PLOS ONE. After reading the reviewer comments and the edits made by the authors, I think the author’s have done a fine job in addressing the concerns brought up by the reviewers and recommend the publication of the manuscript. I do agree with Reviewer 1 in some, but not all aspects of their review. The authors are correct in couching their findings as progressing the laboratory-based research into spot fires and stay within scope throughout the manuscript. The authors acknowledge that their results are limited in inference and that there is much future work to be done to increase the complexity of the analyses and increase the knowledge gained from such experiments.

7. PLOS authors have the option to publish the peer review history of their article (what does this mean?). If published, this will include your full peer review and any attached files.

Reviewer #2: No

---

## [Editor Report · Acceptance letter]

28 Dec 2020

PONE-D-20-18346R1 

Experiments on the influence of spot fire and topography interaction on fire rate of spread 

Dear Dr. Storey:

I'm pleased to inform you that your manuscript has been deemed suitable for publication in PLOS ONE. Congratulations! Your manuscript is now with our production department. 

Kind regards, 

on behalf of

Dr. Min Huang 

Academic Editor

PLOS ONE